# Coagulation profile of COVID-19 patients admitted to the ICU: An exploratory study

Thiago Domingos Corrêa[1,ȣ], Ricardo Luiz Cordioli[1,ȣ]*, João Carlos Campos Guerra[2], Bruno Caldin da Silva[1], Roseny dos Reis Rodrigues[1], Guilherme Martins de Souza[1], Thais Dias Midega[1], Niklas Söderberg Campos[1], Bárbara Vieira Carneiro[1], Flávia Nunes Dias Campos[1], Hélio Penna Guimarães[1], Gustavo Faissol Janot de Matos[1], Valdir Fernandes de Aranda[2], Leonardo José Rolim Ferraz[1]

**1** Department of Critical Care Medicine, Hospital Israelita Albert Einstein, São Paulo, Brazil, **2** Department of Laboratory Center, Hospital Israelita Albert Einstein, São Paulo, Brazil

ȣ These authors contributed equally to this work.
* ricardo.cordioli@einstein.br

**Data Availability Statement:** All relevant data are available from Dryad (doi:10.5061/dryad.pg4f4qrn3).

## Abstract

### Background

Coagulation abnormalities in COVID-19 patients have not been addressed in depth.

### Objective

To perform a longitudinal evaluation of coagulation profile of patients admitted to the ICU with COVID-19.

### Methods

Conventional coagulation tests, rotational thromboelastometry (ROTEM), platelet function, fibrinolysis, antithrombin, protein C and S were measured at days 0, 1, 3, 7 and 14. Based on median total maximum SOFA score, patients were divided in two groups: SOFA $\leq$ 10 and SOFA > 10.

### Results

Thirty patients were studied. Some conventional coagulation tests, as aPTT, PT and INR remained unchanged during the study period, while alterations on others coagulation laboratory tests were detected. Fibrinogen levels were increased in both groups. ROTEM maximum clot firmness increased in both groups from Day 0 to Day 14. Moreover, ROTEM–FIBTEM maximum clot firmness was high in both groups, with a slight decrease from day 0 to day 14 in group SOFA $\leq$ 10 and a slight increase during the same period in group SOFA > 10. Fibrinolysis was low and decreased over time in all groups, with the most pronounced decrease observed in INTEM maximum lysis in group SOFA > 10. Also, D-dimer plasma levels were higher than normal reference range in both groups and free protein S plasma levels were low in both groups at baseline and increased over time, Finally, patients in group SOFA > 10 had lower plasminogen levels and Protein C than patients with SOFA <10, which may represent less fibrinolysis activity during a state of hypercoagulability.

**Funding:** The author(s) received no specific funding for this work.

**Competing interests:** The authors have declared that no competing interests exist.

**Abbreviations:** ARDS, acute respiratory distress syndrome; aPTT, activated partial thromboplastin time; CFT, clot formation time; CCT, conventional coagulation tests; COPD, chronic obstructive pulmonary disease; COVID-19, coronavirus disease 2019; CT, clotting time; DVT, deep vein thrombosis; HFH, unfractionated heparin; ICU, intensive care unit; INR, international normalized ratio; LMWH, low-molecular-weight heparin; LOS, length of stay; MCF, maximum clot firmness; PAI-1, plasminogen activator inhibitor-1; PCC, prothrombin complex concentrate; PT, prothrombin time; RRT, renal replacement therapy; ROTEM, rotational thromboelastometry; RT-PCR, reverse-transcriptase-polymerasechain-reaction; SAPS III, simplified acute physiology score III; SARS-CoV-2, severe acute respiratory syndrome coronavirus 2; SOFA, sequential organ failure assessment score; TT, thrombin time; UFH, unfractionated heparin.

## Conclusion

COVID-19 patients have a pronounced hypercoagulability state, characterized by impaired endogenous anticoagulation and decreased fibrinolysis. The magnitude of coagulation abnormalities seems to correlate with the severity of organ dysfunction. The hypercoagulability state of COVID-19 patients was not only detected by ROTEM but it much more complex, where changes were observed on the fibrinolytic and endogenous anticoagulation system.

## Introduction

More than thirteen million people have been diagnosed with coronavirus disease 2019 (COVID-19) worldwide [1] since severe acute respiratory syndrome coronavirus 2 (SARS-CoV-2) was first identified in China in December 2019 [2]. Almost one third of the hospitalized patients with COVID-19 are admitted to the ICU [2–4].

Coagulation abnormalities, mainly thrombotic complications, have been described in COVID-19 patients [5–11]. The incidence of arterial and venous thrombotic complications in COVID-19 patients admitted to the ICU may reach 31% [12]. Indeed, it has been shown that D-dimer and fibrinogen degradation products (FDP) values were higher in patients with most severe SARS-CoV-2 infection than in patients with milder forms of disease [6]. Moreover, Chen et al. have demonstrated that deceased patients with COVID-19 exhibited D-dimer concentrations approximately seven times higher than recovered patients [13]. Additionally, low platelet count was show to be associated with increased risks of more severe forms of disease and increased hospital mortality in COVID-19 patients [10]. Although D-dimer, FDP and platelet count have been used as clinical indicators of SARS-CoV-2 infection severity [14], coagulation abnormalities in COVID-19 patients have not been addressed in depth.

Conventional coagulation tests (CCT) such as prothrombin time (PT), international normalized ratio (INR), thrombin time (TT) and activated partial thromboplastin time (aPTT) are unable to reflect the complexity of hemostatic impairment observed in intensive care unit (ICU) patients [15]. Rotational thromboelastometry (ROTEM) represents a point-of-care test that assess the viscoelastic properties of whole blood, providing a real-time evaluation of clot formation kinetics, i.e., clot formation, stabilization and dissolution, at the bedside [16]. To the best of our knowledge, a comprehensive and longitudinal analysis of the coagulation profile, including CCT, ROTEM, platelet function, fibrinolysis, and endogenous inhibitors of coagulation (antithrombin, protein C and S) have not been described in ICU COVID-19 patients so far.

We hypothesized that ICU patients diagnosed with COVID-19 have a prothrombotic profile that may be explained by an impaired endogenous anticoagulation system. Moreover, the degree of coagulation abnormalities reflects the severity of the disease. Therefore, our objective was to perform a comprehensive longitudinal evaluation of coagulation, fibrinolysis, and endogenous anticoagulation system of patients admitted to the ICU with severe COVID-19.

## Materials and methods

### Study design and setting

We performed a single center prospective longitudinal study in an ICU of a private tertiary care hospital in São Paulo, Brazil. The study was approved by the Local Ethics Committee at

Hospital Israelita Albert Einstein and by Comissão Nacional de Ética em Pesquisa (CONEP) with waiver of informed consent (CAAE: 30175220.3.0000.0071). This study is reported in accordance with the Strengthening the Reporting of Observational studies in Epidemiology (STROBE) statement [17].

## Study participants

Thirty patients aged ≥18 years old admitted to the ICU with confirmed diagnosis of COVID-19 were included in this study. Laboratory confirmation of SARS-CoV-2 infection was based on positive reverse-transcriptase-polymerasechain-reaction (RT-PCR) assay [18].

Exclusion criteria included pregnancy, previous known coagulopathy, currently use of systemic anticoagulants or anti-platelet therapy or vitamin K antagonists, moribund patients and patients who presented cardiac arrest.

Participants were recruited between March 29, 2020 through May 13, 2020 and they could represent the majority of severe patients infected by COVID-19, once there were few exclusion criteria and the participants were recruited with waiver of informed consent once it was an observational study without any intervention and consecutive patients admitted in the Intensive Care Unit were recruited following the inclusion and exclusion criteria until we completed 30 patients included in the study.

## Laboratory analysis

Laboratory tests were performed at the time of study inclusion (baseline), and at days 1, 3, 7 and 14 after enrollment unless the patient had died or was discharged from the hospital.

**Conventional coagulation tests.** Conventional coagulation tests included: platelet count (XE 2100, Sysmex, São Paulo, Brazil), plasma fibrinogen concentration [Clauss method (Hemosil QFA thrombin (bovine), IL Instrumentation Laboratory Company, Bedford MA, USA], aPTT (Hemosil Synthasil, IL Instrumentation Laboratory Company, Bedford MA, USA), PT and INR (Hemosil PT-Fibrinogen HS Plus, IL Instrumentation Laboratory Company, Bedford MA, USA) and ionic calcium (ABL 800 FLEX, Radiometer Medical ApS, Brønshøj, Denmark).

**Rotational thromboelastometry.** Rotational thromboelastometry analyses were performed with EXTEM (extrinsic coagulation pathway assessment), INTEM (intrinsic coagulation pathway assessment) and FIBTEM (extrinsic coagulation pathway assessment with additional platelet inhibition using cytochalasin D) tests according to the manufacturer's instructions [16]. The following parameters were recorded during ROTEM analysis: clotting time [CT; seconds (sec)], which represents the beginning of the test until clot firmness of 2 mm; clot formation time (CFT; sec), which represents time between detection of a clot firmness of 2 and 20 mm and maximum clot firmness (MCF; mm), which represents the greatest amplitude of thromboelastometric trace and reflects clot "strength" [19, 20]. ROTEM tests were performed by laboratory technicians. Blood samples of approximately 3 ml were collected by venipuncture into a tube with citrate (3.2%; Sarsted1, Wedel, Germany). Blood samples were immediately processed for ROTEM analysis. The analyses were performed by pipetting 340 μl of citrated whole blood and 20 μl of 0.2 M calcium chloride with specific activators into a cup. There was no change in methodology for test performance nor test controls (Rotrol N and Rotrol P) throughout the study period [15, 20].

Normal coagulation profile on the ROTEM was defined according to reference values for CT, CFT and MCF (INTEM CT: 100–240 sec, INTEM CFT: 30–110 sec, INTEM MCF: 50–72 mm; EXTEM CT: 38–79 sec, EXTEM CFT: 34–159 sec, EXTEM MCF: 50–72 mm; FIBTEM MCF: 9–25 mm) [15, 19, 20]. Hypocoagulability in ROTEM was defined as prolongation of

CT (INTEM CT >240 sec or EXTEM CT >79 sec) and/or CFT (INTEM CFT >110 sec or EXTEM CFT >159 sec) and/or MCF reduction (MCF INTEM or EXTEM MCF <50 mm or FIBTEM MCF <9 mm) [15, 19]. Hypercoagulability in ROTEM was defined as a reduction in clotting time (INTEM CT <100 sec or EXTEM CT <38 sec), or clot formation time (INTEM CFT <30 sec or EXTEM CFT <34 sec) and/or an increase in MCF (MCF INTEM or EXTEM MCF >72 mm or FIBTEM MCF >25 mm) [15, 19].

**Platelet function test.** Platelet function test was assessed in whole blood samples using impedance aggregometry (The ROTEM®-Platelet; TEM Innovations GmbH, Munich Germany) [21]. Platelets were activated with arachidonic acid (ARATEM test) and adenosine diphosphate (ADPTEM) [21].

**Fibrinolysis and endogenous anticoagulation system.** D-dimer (Hemosil D-dimer HS 500 and hemosil D-dimer HS 500 controls, L Instrumentation Laboratory Company, Bedford MA, USA), serum plasminogen (Plasmin Inhibitor, IL Instrumentation Laboratory Company, Bedford MA, USA), alpha-2 antiplasmin (Plasmin Inhibitor, IL Instrumentation Laboratory Company, Bedford MA, USA), antithrombin (IL Instrumentation Laboratory Company, Bedford MA, USA), protein C (IL Instrumentation Laboratory Company, Bedford MA, USA) and free protein S (IL Instrumentation Laboratory Company, Bedford MA, USA) were measured.

## Data collection

All study data were retrieved from Epimed Monitor System (Epimed Solutions, Rio de Janeiro, Brazil), which is an electronic structured case report form where patients data are prospectively entered by trained ICU case managers [22].

Collected clinical variables included demographics, comorbidities, Simplified Acute Physiology score (SAPS 3 score) at ICU admission [23], Sequential Organ Failure Assessment score (SOFA score) [24] at ICU admission, and at days 1, 3, 7 and 14 after enrollment unless the patient had died or was discharged from the ICU, total maximum SOFA score (from the time of study inclusion (baseline) up to 14 after enrollment unless the patient had died or was discharged from the ICU) [25], body-mass index, treatment measures (i.e, hydroxychloroquine, macrolides, corticosteroids, interleukin-6 receptor antagonist, convalescent plasma and lopinavir-ritonavir), supportive therapy [use of vasopressors, mechanical ventilation, noninvasive mechanical ventilation and renal replacement therapy (RRT)] during ICU stay, hospital length of stay (LOS) prior to ICU admission, ICU and hospital LOS, and ICU mortality.

Blood component transfusion [platelet concentrate, fresh frozen plasma (FFP) and cryoprecipitate] and hemostatic agents [fibrinogen concentrate, prothrombin complex concentrate (PCC) and tranexamic acid] were collected. The presence of thrombotic or hemorrhagic events and the use of prophylactic or therapeutic doses of low-molecular-weight heparin (LMWH) or unfractionated heparin (UFH) during ICU stay were recorded.

## Statistical analysis

Based on median total maximum SOFA score, patients were divided in two groups: group SOFA ≤ 10 and group SOFA > 10. Categorical variables were presented as n/n total (%). Continuous variables were presented as median with interquartile range (IQR). Categorical variables were compared between groups with Fisher's exact test, and continuous variables were compared using independent t test or Mann-Whitney U test in case of non-normal distribution, tested by the Kolmogorov-Smirnov test.

To account for longitudinal (repeated measurements) and correlated response continuous variables, between-group differences and within-group differences over time were assessed using generalized estimating equations (GEE), with group (SOFA ≤ 10 and group

SOFA > 10) and study time points (time) as predictors. P values for group effect, time effect, and time-group interaction were presented. When a time effect was detected in pooled patients, each time point (Day 1, 3, 7 and 14) was compared against Day 0. When a group effect or a time-group interaction were detected, between group comparisons (group SOFA>10 vs. group SOFA ≤10) were performed at each time point. The Bonferroni method was used to account for multiple comparisons.

Two-tailed tests were used and when p<0.05, the test was considered statistically significant. No adjustment was made for missing data. The SPSS™ (IBM™ Statistical Package for the Social Science version 26.0) was used for statistical analyses, and GraphPad Prism version 8.0.0 (GraphPad Software, San Diego, California, USA) was used for graph plotting.

## Results

### Baseline characteristics of patients

From March 29, 2020 through May 13, 2020, thirty patients were included in this study. The median (IQR) age of pooled patients was 61 (52–83) years, 50.0% were man and median (IQR) SAPS III of 49 (41–61). Of the 30 patients, 24 (80.0%) had one or more coexisting medical conditions. Obesity [12/29 (41.4%)], systemic hypertension [12/30 (40.0%)] and diabetes mellitus [11/30 (36.7%)] were the most common coexisting conditions. Most patients received invasive mechanical ventilation [27/30 (90.0%)] and/or vasopressors [27/30 (90.0%)] during the ICU stay. Clinical characteristics of patients are shown in Table 1.

Compared with patients in group SOFA ≤ 10 [16/30 (53.3%) patients], patients in group SOFA > 10 [14/30 (46.7%)] were older [median (IQR), 78 (60–85) vs. 53 (45–64) years, p = 0.002], had a higher SOFA score at study inclusion [median (IQR), 8 (6–9) vs. 5 (3–6) points, p = 0.002], a higher number of coexisting conditions [median (IQR), 3 (2–3) vs. 1 (0–2), p<0.001], had diabetes mellitus more frequently [9/14 (64.3) vs. 2/16 (12.5), p = 0.007] and received renal replacement therapy more frequently [10/14 (71.4) vs. 0/16 (0.0), p<0.001] (Table 1).

### Anticoagulants, blood transfusion and clinical outcomes

During the study period, 22/30 (73.3%) patients received anticoagulants as deep vein thrombosis (DVT) prophylaxis and 7/30 (23.3%) patients received systemic anticoagulation (Table 2). The proportion of patients receiving anticoagulants as DVT prophylaxis and as systemic anticoagulation did not differ between the groups (p = 0.830) (Table 2). Unfractionated heparin was more commonly used as DVT in group SOFA > 10 compared to group SOFA ≤ 10 [7/10 (70.0%) vs. 2/12 (16.7%), p = 0.027] (Table 2).

Four patients (13.3%) received red blood cells transfusion, all of them in group SOFA > 10 (Table 2). No patient received platelet concentrate, FFP, cryoprecipitate, fibrinogen concentrate, PCC or tranexamic acid during the study period. Thrombotic events occurred in 6/30 (20.0%) patients and hemorrhagic events were observed in 3/30 (10.0%) patients. The incidence of thrombotic and hemorrhagic events did not differ between the groups (Table 2).

As of June 10, 2020, one patient [1/30 (3.3%)] was still hospitalized in the ICU while 25/30 (83.3%) of patients were discharged alive from the ICU. Four [4/30 (13.3%)] patients died at the ICU (Table 2). Compared with patients in group SOFA ≤ 10, patients in group SOFA > 10 had a higher ICU mortality [4/14 (28.6%) vs. 0/16 (0.0%), p = 0.014], exhibited a higher ICU [median (IQR), 22 (16–35) vs. 7 (6–15), p<0.001], and hospital [38 (25–51) vs. 17 (13–30), p = 0.012] LOS (Table 2).

**Table 1. Characteristics of study participants.**

| Characteristics | All patients (n = 30) | SOFA ≤10 (n = 16) | SOFA >10 (n = 14) | P value |
|---|---|---|---|---|
| Age, years (median, IQR) | 61 (52–83) | 53 (45–64) | 78 (60–85) | 0.002[a] |
| Men, n (%) | 15/30 (50.0) | 7/16 (43.8) | 8/14 (57.1) | 0.715[b] |
| SAPS III score, points (median, IQR)[§] | 49 (41–61) | 50 (43–64) | 47 (41–55) | 0.307[a] |
| SOFA score D0, points (median, IQR)[#] | 6 (4–8) | 5 (3–6) | 8 (6–9) | 0.002[a] |
| Maximum SOFA score, points (median, IQR)[#] | 10 (7–12) | 7 (6–9) | 13 (11–14) | <0.001[a] |
| Number of coexisting conditions, (median, IQR) | 2 (1–3) | 1 (0–2) | 3 (2–3) | <0.001[a] |
| Coexisting conditions, n (%) | | | | |
| Obesity | 12/29 (41.4) | 7/16 (43.8) | 5/13 (38.5) | 1.000[b] |
| Systemic hypertension | 12/30 (40.0) | 5/16 (31.3) | 7/14 (50.0) | 0.457[b] |
| Diabetes mellitus | 11/30 (36.7) | 2/16 (12.5) | 9/14 (64.3) | 0.007[b] |
| Malignancy | 4/30 (13.3) | 2/16 (12.5) | 2/14 (14.3) | 1.000[b] |
| Congestive heart failure | 3/30 (10.0) | 0/16 (0.0) | 3/14 (21.4) | 0.090[b] |
| COPD / Asthma | 3/30 (10.0) | 1/16 (6.3) | 2/14 (14.3) | 0.586[b] |
| Chronic kidney disease | 4/30 (13.3) | 1/16 (6.3) | 3/14 (21.4) | 0.315[b] |
| Coronary artery disease | 2/30 (6.7) | 0/16 (0.0) | 2/14 (14.3) | 0.209[b] |
| Body-mass index | 29.3 (24.4–32.2) | 29.7 (24.2–32.5) | 27.2 (24.4–31.4) | 0.793[a] |
| Treatment, n (%) | | | | |
| Macrolides | 28/30 (93.3) | 16/16 (100.0) | 12/14 (85.7) | 0.209[b] |
| Glucocorticoids | 25/30 (83.3) | 12/16 (75.0) | 13/14 (92.9) | 0.336[b] |
| Hydroxychloroquine | 24/30 (80.0) | 13/16 (81.3) | 11/14 (78.6) | 1.000[b] |
| Convalescent plasma | 10/30 (33.3) | 6/16 (37.5) | 4/14 (28.6) | 0.709[b] |
| Interleukin-6 receptor antagonist | 3/30 (10.0) | 2/16 (12.5) | 1/14 (7.1) | 1.000[b] |
| Lopinavir-ritonavir | 2/30 (6.7) | 1/16 (6.3) | 1/14 (7.1) | 1.000[b] |
| Support during ICU stay, n (%) | | | | |
| Vasopressors | 27/30 (90.0) | 13/16 (81.3) | 14/14 (100.0) | 0.228[b] |
| Mechanical ventilation | 27/30 (90.0) | 13/16 (81.3) | 14/14 (100.0) | 0.228[b] |
| Noninvasive ventilation | 6/30 (20.0) | 3/16 (18.8) | 3/14 (21.4) | 1.000[b] |
| Renal replacement therapy | 10/30 (33.3) | 0/16 (100.0) | 10/14 (71.4) | <0.001[b] |
| Time from symptom onset to study inclusion, days (median, IQR) | 9 (6–14) | 9 (6–15) | 8 (6–13) | 0.951[c] |
| Hospital LOS prior ICU admission, days (median, IQR) | 1 (0–3) | 2 (1–3) | 1 (0–2) | 0.400[c] |

Values represent median (IQR) or n (%). ICU: intensive care unit, SAPS III: simplified acute physiology score III

§: scores on SAPS III range from 0 to 217, with higher scores indicating more severe illness and higher risk of death, SOFA: sequential organ failure assessment score

#: SOFA score ranges from 0 to 24, with higher scores indicating more severe organ dysfunction, COPD: chronic obstructive pulmonary disease, LOS: length of stay. P values were calculated with the use of (a) Independent t-test, (b) Fisher's exact test or (c) Mann-Whitney U test.

## Laboratory analysis

Arterial pH increased in both groups from Day 0 to Day 14 (p<0.001 for time effect), while ionized calcium increased from Day 0 to Day 14 in group SOFA ≤ 10, and decreased in group SOFA > 10 (p = 0.038 for time-group interaction) (Table 3). Peripheral temperature remained stable during study period (Table 3).

**Conventional coagulation tests.** Platelet count increased from baseline to Day 14 in both groups (p<0.001 for time effect), although lower platelet count was observed over the time in group SOFA > 10 compared to group SOFA ≤ 10 (p = 0.009 for group effect) (Table 3). Fibrinogen levels were increased in both groups at baseline, with the highest values observed at day 1 in group SOFA ≤ 10 and at day 3 in group SOFA > 10 (p<0.001 for time effect). Patients in group SOFA ≤ 10 exhibited a more pronounced decrease in plasma fibrinogen levels at

**Table 2. Administered treatments and clinical outcomes.**

| Characteristics | All patients (n = 30) | SOFA ≤10 (n = 16) | SOFA >10 (n = 14) | P value |
|---|---|---|---|---|
| Anticoagulants, n (%) | | | | 0.830[a] |
| DVT prophylaxis | 22/30 (73.3) | 12/16 (75.0) | 10/14 (71.4) | |
| UFH | 9/22 (40.9) | 2/12 (16.7) | 7/10 (70.0) | 0.027[a] |
| LMWH | 13/22 (59.1) | 10/12 (83.3) | 3/10 (30.0) | |
| Systemic anticoagulation | 7/30 (23.3) | 3/16 (18.8) | 4/14 (28.6) | |
| UFH | 5/7 (71.4) | 1/3 (33.3) | 4/4 (100.0) | 0.143[a] |
| LMWH | 2/7 (28.6) | 2/3 (66.7) | 0/4 (0.0) | |
| Transfused blood product, n (%) | | | | |
| Red blood cells | 4/30 (13.3) | 0 (0.0) | 4/14 (28.6) | 0.037[a] |
| Platelet concentrate | 0/30 (0.0) | 0 (0.0) | 0 (0.0) | |
| Fresh frozen plasma | 0/30 (0.0) | 0 (0.0) | 0 (0.0) | |
| Cryoprecipitate | 0/30 (0.0) | 0 (0.0) | 0 (0.0) | |
| Thrombotic events, n (%) | 6/30 (20.0) | 3/16 (18.8) | 3/14 (21.4) | 1.000[a] |
| DVT | 4/6 (66.7) | 2/3 (66.7) | 2/3 (66.7) | |
| Pulmonary embolism | 2/6 (33.3) | 1/3 (33.3) | 1/3 (33.3) | |
| Hemorrhagic events, n (%) | 3/30 (10.0) | 2/16 (12.5) | 1/14 (7.1) | 1.000[a] |
| Outcomes[#] | | | | |
| Died in ICU | 4/30 (13.3) | 0/16 (0.0) | 4/14 (28.6) | 0.014[a] |
| Discharged from ICU | 25/30 (83.3) | 16/16 (100.0) | 9/14 (64.3) | |
| Still in ICU as of 06/10/2020 | 1/30 (3.3) | 0/16 (0.0) | 1/14 (7.1) | |
| ICU LOS (days), median (IQR)[#] | 15 (7–22) | 7 (6–15) | 22 (16–35) | <0.001[b] |
| Hospital LOS (days), median (IQR)[#] | 26 (15–38) | 17 (13–30) | 38 (25–51) | 0.012[b] |

Values represent median (IQR) or n (%). ICU: intensive care unit, DVT: deep vein thrombosis, HFH: unfractionated heparin, LMWH: low-molecular-weight heparin, LOS: length of stay.

#: the number of patients who died, were discharged, and were still admitted in the ICU as of June 10, 2020 were recorded, and ICU and hospital length of stay also were determined. P values were calculated with the use of (a) Fisher's exact test and (b) Mann-Whitney U test.

study end than patients in group SOFA > 10 (p = 0.048 for time-group interaction) (Table 3). Prothrombin time, INR, aPTT, and hemoglobin remained unchanged during the study period (Table 3).

**Rotational thromboelastometry.** The majority of patients in both groups exhibited a hypercoagulability state based on ROTEM (Fig 1). ROTEM (INTEM and EXTEM) maximum clot firmness slightly increased in both study groups from Day 0 to Day 14 (p<0.001 for time effect) (Table 4). ROTEM–FIBTEM maximum clot firmness was high in both groups during the study period, with a slight decrease from Day 0 to Day 14 in group SOFA ≤ 10 and a slight increase during the same period in group SOFA > 10 (p = 0.050 for time-group interaction) (Table 4).

**Platelet function test.** Median (IQR) values of ARATEM and ADPTEM tests remained within the normal range during the study period, although both ARATEM (p = 0.014 for time effect) and ADPTEM (p = 0.004 for time effect) slightly increased over time in both groups (Table 4).

**Fibrinolysis and endogenous inhibitors of coagulation.** Fibrinolysis (INTEM and EXTEM maximum lysis) was low and decreased over time in all groups, with the most pronounced decrease observed in INTEM maximum lysis in group SOFA > 10 (p = 0.004 for time-group interaction) (Table 4). D-dimer plasma levels were 2 to 3-fold higher than normal reference range in both groups (Fig 2; S1 Table). Plasminogen median values remained within

**Table 3. Arterial pH, ionized calcium, peripheral temperature, conventional coagulation tests and hemoglobin.**

| Parameters | Reference range | Day 0 | Day 1 | Day 3 | Day 7 | Day 14 | P value |
|---|---|---|---|---|---|---|---|
| Arterial pH | 7.35–7.45 | | | | | | |
| All patients | | 7.40 (7.35–7.41) | 7.38 (7.31–7.40) | 7.40 (7.34–7.43) | 7.42 (7.38–7.45) | 7.46 (7.43–7.50)* | <0.001[a] |
| SOFA≤10 | | 7.40 (7.36–7.41) | 7.39 (7.35–7.42) | 7.43 (7.40–7.45) | 7.43 (7.42–7.47) | 7.45 (7.42–7.48) | <0.001[b] |
| SOFA >10 | | 7.39 (7.31–7.40) | 7.32 (7.27–7.39)# | 7.34 (7.30–7.39)# | 7.41 (7.32–7.45)# | 7.47 (7.44–7.50) | 0.001[c] |
| Ionized calcium (mmol/L) | 1.14–1.31 | | | | | | |
| All patients | | 1.12 (1.09–1.15) | 1.15 (1.12–1.20) | 1.13 (1.10–1.16) | 1.17 (1.15–1.21) | 1.15 (1.06–1.24) | 0.030[a] |
| SOFA≤10 | | 1.14 (1.13–1.21) | 1.15 (1.12–1.19) | 1.13 (1.10–1.14) | 1.16 (1.15–1.19) | 1.24 (1.17–1.25) | 0.151[b] |
| SOFA >10 | | 1.12 (1.09–1.12)# | 1.14 (1.10–1.20) | 1.13 (1.09–1.21) | 1.19 (1.16–1.26) | 1.09 (1.06–1.17) | 0.038[c] |
| Temperature (˚C) | | | | | | | |
| All patients | | 36.4 (36.0–37.1) | 36.6 (36.2–37.0) | 36.5 (36.2–37.1) | 36.2 (36.0–36.7) | 36.1 (36.0–36.6) | 0.149[a] |
| SOFA≤10 | | 36.2 (36.0–36.8) | 36.6 (36.4–37.2) | 36.7 (36.2–37.2) | 36.3 (36.2–36.7) | 36.1 (36.0–36.6) | 0.351[b] |
| SOFA >10 | | 36.5 (35.9–37.5) | 36.5 (36.0–37.0) | 36.3 (35.3–36.7) | 36.1 (35.9–36.8) | 36.1 (35.7–36.6) | 0.324[c] |
| Platelets (x10$^9$/L) | 150–450 | | | | | | |
| All patients | | 226 (176–261) | 236 (182–268) | 272 (230–314)* | 349 (242–444)* | 306 (229–480)* | <0.001[a] |
| SOFA≤10 | | 243 (185–274) | 236 (210–307) | 271 (252–310) | 419 (307–462) | 469 (232–679) | 0.009[b] |
| SOFA >10 | | 197 (141–237)# | 220 (160–264) | 277 (197–314) | 282 (175–349)# | 286 (212–383) | 0.213[c] |
| Fibrinogen (g/dL) | 200–400 | | | | | | |
| All patients | | 600 (480–680) | 642 (470–722) | 625 (513–782) | 532 (348–592)* | 397 (303–537)* | <0.001 |
| SOFA≤10 | | 633 (503–690) | 642 (524–722) | 592 (513–680) | 482 (348–592) | 372 (298–439) | 0.365[b] |
| SOFA >10 | | 552 (480–680) | 610 (470–851) | 700 (513–822) | 564 (473–646) | 470 (303–550) | 0.048[c] |
| Prothrombin time (sec) | 70–100 | | | | | | |
| All patients | | 82 (76–89) | 81 (69–89) | 78 (70–88) | 85 (73–88) | 80 (67–89) | 0.323[a] |
| SOFA≤10 | | 82 (78–93) | 78 (70–86) | 78 (70–88) | 85 (78–87) | 83 (67–97) | 0.226[b] |
| SOFA >10 | | 83 (67–86) | 86 (69–89) | 75 (67–86) | 76 (70–89) | 77 (63–87) | 0.463[c] |
| INR | 0.96–1.30 | | | | | | |
| All patients | | 1.13 (1.07–1.18) | 1.14 (1.07–1.25) | 1.17 (1.10–1.25) | 1.11 (1.07–1.21) | 1.17 (1.07–1.34) | 0.503[a] |
| SOFA≤10 | | 1.13 (1.05–1.17) | 1.16 (1.09–1.25) | 1.16 (1.07–1.24) | 1.10 (1.09–1.16) | 1.12 (1.02–1.33) | 0.303[b] |
| SOFA >10 | | 1.13 (1.09–1.28) | 1.10 (1.07–1.26) | 1.19 (1.10–1.29) | 1.18 (1.07–1.27) | 1.21 (1.10–1.39) | 0.143[c] |
| aPTT (sec) | 25.6–35.5 | | | | | | |
| All patients | | 28.8 (27.2–32.6) | 27.8 (25.6–32.8) | 28.6 (25.9–33.2) | 27.9 (26.4–31.9) | 28.1 (25.6–35.7) | 0.079[a] |
| SOFA≤10 | | 28.0 (27.0–31.1) | 27.2 (25.6–31.2) | 26.7 (25.6–31.0) | 27.4 (27.0–30.8) | 28.1 (23.7–31.9) | 0.497[b] |
| SOFA >10 | | 30.0 (28.3–33.0) | 28.7 (25.6–33.4) | 30.8 (27.0–38.0) | 28.4 (26.0–33.3) | 28.3 (26.6–36.7) | 0.512[c] |
| Hemoglobin (g/dL) | 13.5–17.5 | | | | | | |
| All patients | | 12.1 (11.2–12.9) | 11.4 (10.2–12.2)* | 10.5 (9.6–11.8)* | 10.2 (9.3–11.1)* | 9.2 (8.7–10.6)* | <0.001[a] |
| SOFA≤10 | | 12.2 (11.3–12.9) | 11.4 (10.7–12.1) | 10.6 (9.7–11.8) | 10.7 (10.1–11.1) | 9.2 (9.1–11.7) | 0.483[b] |
| SOFA >10 | | 11.7 (11.2–13.4) | 11.3 (10.1–13.2) | 10.5 (9.0–11.8) | 9.8 (8.3–10.3) | 9.3 (8.1–10.2) | 0.618[c] |

Values represent median (IQR). INR: international normalized ratio, aPTT: activated partial thromboplastin time. P values were calculated with the use of generalized estimating equations (GEE): (a): time effect, (b): group effect and (c): time-group interaction. Pairwise comparisons significant at the 0.05 level:

(*): time effect—pooled patients: each time point vs. Day 0.

(#): between group comparisons (group SOFA>10 vs. group SOFA ≤10) at each time point.

the normal range during the study period, although with a slight increase over time in both groups (p<0.001 for time effect). Alpha-2 antiplasmin remained unchanged during the study period (Fig 2; S1 Table).

Antithrombin slightly increased over time in group SOFA ≤ 10 while it remained stable in group SOFA > 10 (p = 0.021 for time-group interaction) (Fig 2; S1 Table). Protein C plasma levels increased over time in both groups, although patients in group SOFA > 10 exhibited

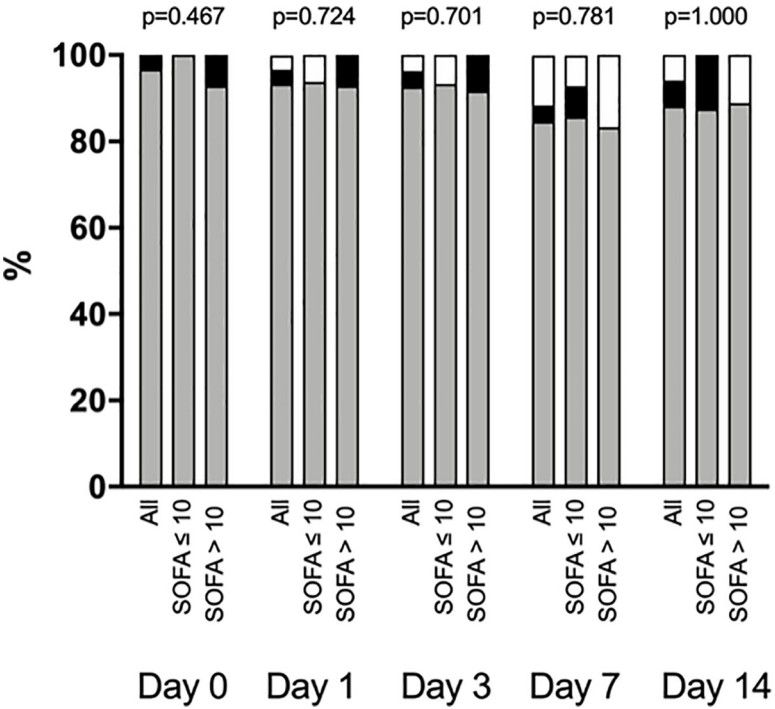

**Fig 1. Coagulation profile accordingly to rotational thromboelastometry (ROTEM).** White filled bars represent normal coagulation profile, grey filled bars represent a hypercoagulability state and black filled bars represent a hypocoagulability state. P values comparing group SOFA>10 vs. group SOFA ≤10 at each time point were calculated with the use of Fisher's exact test.

lower values in comparison to patients in group SOFA ≤ 10 (p = 0.015 for time-group interaction) (Fig 2; S1 Table). Free Protein S plasma levels were low in both groups at baseline and increased over time with no between-group differences (p<0.001 for time effect) (Fig 2; S1 Table).

## Discussion

In this prospective longitudinal single center study, we demonstrated that patients admitted to the ICU with severe SARS-CoV-2 infection exhibited a pronounced hypercoagulability state, characterized by increased plasma fibrinogen levels, decreased free protein S plasma levels, and decreased fibrinolysis. The severity of coagulation derangements seems to correlate with the intensity of organ dysfunction according to the SOFA score. Finally, the hypercoagulability state of severe COVID-19 patients was detected by ROTEM and modifications on some coagulation tests related to the fibrinolytic and endogenous anticoagulation system, while the more common conventional coagulation tests, as aPTT (sec) INR and platelets, remained unchanged.

Approximately 5% of patients infected with SARS-CoV-2 become critically ill, developing organ dysfunction and failure [26]. Coagulopathy is frequently observed in COVID-19 patients [5–11] and has been associated with worse outcomes [3, 11, 27, 28]. For instance, Tang *and cols.* showed that abnormal conventional coagulation tests, especially markedly elevated D-dimer and FDP levels during hospitalization, were associated with poor prognosis in COVID-19 patients [11]. Moreover, disseminated intravascular coagulation (DIC) was more frequently observed in non-survivals (71.4% vs. 0.6%, respectively) compared to survivors [11].

**Table 4. Rotational thromboelastometry and platelet function test.**

| Parameters | Reference range | Day 0 | Day 1 | Day 3 | Day 7 | Day 14 | P value |
|---|---|---|---|---|---|---|---|
| **ROTEM—INTEM** | | | | | | | |
| Clotting time (sec) | 100–240 | | | | | | |
| All patients | | 164 (155–184) | 160 (156–191) | 173 (159–196) | 171 (157–193) | 163 (153–190) | 0.300[a] |
| SOFA≤10 | | 165 (141–180) | 157 (153–168) | 161 (142–180) | 172 (162–193) | 171 (148–188) | 0.035[b] |
| SOFA >10 | | 159 (157–184) | 176 (156–194) | 184 (172–214)[#] | 169 (152–185) | 157 (153–194) | 0.378[c] |
| Clot formation time (sec) | 30–110 | | | | | | |
| All patients | | 48 (46–59) | 47 (42–57) | 44 (39–55) | 42 (39–48) | 41 (34–47) | 0.096[a] |
| SOFA≤10 | | 48 (45–55) | 46 (41–52) | 40 (35–47) | 42 (37–47) | 34 (29–49) | 0.026[b] |
| SOFA >10 | | 53 (47–62) | 51 (43–63)[#] | 52 (45–62)[#] | 43 (40–61) | 43 (39–46) | 0.567[c] |
| Maximum clot firmness (mm) | 50–72 | | | | | | |
| All patients | | 70 (67–72) | 71 (69–73)[*] | 73 (70–76) | 75 (70–78)[*] | 76 (69–79)[*] | <0.001[a] |
| SOFA≤10 | | 70 (68–72) | 72 (70–74) | 74 (72–76) | 75 (70–79) | 78 (70–80) | 0.312[b] |
| SOFA >10 | | 70 (66–73) | 71 (67–73) | 72 (70–76) | 75 (71–77) | 75 (69–78) | 0.914[c] |
| Maximum lysis (%) | £15 | | | | | | |
| All patients | | 10 (6–12) | 8 (5–10)[*] | 6 (3–9)[*] | 3 (2–6)[*] | 4 (2–6)[*] | <0.001[a] |
| SOFA≤10 | | 11 (6–13) | 8 (4–10) | 7 (6–9) | 5 (2–6) | 5 (2–6) | 0.223[b] |
| SOFA >10 | | 9 (6–12) | 8 (5–11) | 3 (2–7) | 2 (1–3)[#] | 3 (2–7) | 0.004[c] |
| **ROTEM—EXTEM** | | | | | | | |
| Clotting time (sec) | 38–79 | | | | | | |
| All patients | | 72 (66–79) | 73 (66–88) | 78 (70–84)[*] | 73 (62–88) | 72 (61–80) | 0.013[a] |
| SOFA≤10 | | 76 (71–81) | 75 (67–94) | 76 (68–78) | 74 (68–81) | 78 (61–83) | 0.486[b] |
| SOFA >10 | | 68 (66–71) | 72 (64–75) | 85 (77–94)[#] | 68 (61–89) | 67 (65–80) | 0.001[c] |
| Clot formation time (sec) | 34–159 | | | | | | |
| All patients | | 54 (44–61) | 56 (45–64) | 48 (41–63) | 49 (41–57) | 47 (41–56) | 0.123[a] |
| SOFA≤10 | | 53 (44–61) | 52 (43–62) | 45 (40–64) | 46 (37–56) | 38 (31–59) | 0.124[b] |
| SOFA >10 | | 54 (50–64) | 60 (45–73) | 56 (47–63) | 53 (45–62) | 51 (44–55) | 0.839[c] |
| Maximum clot firmness (mm) | 50–72 | | | | | | |
| All patients | | 73 (69–74) | 73 (70–75) | 75 (71–77) | 74 (70–79)[*] | 74 (69–79) | <0.001[a] |
| SOFA≤10 | | 73 (70–75) | 74 (71–76) | 76 (71–77) | 75 (70–79) | 75 (69–80) | 0.344[b] |
| SOFA >10 | | 71 (68–73) | 72 (68–73) | 73 (71–80) | 74 (72–78) | 74 (69–78) | 0.415[c] |
| Maximum lysis (%) | £15 | | | | | | |
| All patients | | 10 (7–12) | 9 (6–13) | 8 (5–10)[*] | 8 (3–11) | 7 (5–11) | 0.006[a] |
| SOFA≤10 | | 10 (8–12) | 9 (6–11) | 8 (6–9) | 8 (4–12) | 7 (4–11) | 0.863[b] |
| SOFA >10 | | 9 (6–15) | 10 (7–13) | 8 (4–11) | 4 (3–10) | 8 (5–11) | 0.617[c] |
| **ROTEM—FIBTEM** | | | | | | | |
| Maximum clot firmness (mm) | 9–25 | | | | | | |
| All patients | | 36 (32–38) | 37 (30–40) | 41(30–44) | 37 (31–45) | 35 (28–47) | 0.080[a] |
| SOFA≤10 | | 37 (30–41) | 37 (30–41) | 40 (28–43) | 42 (31–46) | 32 (27–49) | 0.836[b] |
| SOFA >10 | | 36 (33–38) | 36 (31–40) | 42 (36–46) | 37 (32–42) | 38 (31–43) | 0.050[c] |
| **PLATELET function test** | | | | | | | |
| ARATEM test (sec) | 70–153 | | | | | | |
| All patients | | 79 (54–110) | 87 (58–113) | 81 (62–112) | 114 (89–124)[*] | 110 (83–154)[*] | 0.014[a] |
| SOFA≤10 | | 79 (53–108) | 88 (65–112) | 85 (71–110) | 121 (115–154) | 140 (85–182) | 0.068[b] |
| SOFA >10 | | 79 (54–122) | 76 (36–113) | 62 (43–112) | 95 (65–110)[#] | 109 (83–116) | 0.161[c] |
| ADPTEM test (sec) | 56–139 | | | | | | |
| All patients | | 96 (65–111) | 105 (72–128) | 92 (76–117) | 127 (86–138) | 112 (67–134) | 0.004[a] |
| SOFA≤10 | | 105 (83–126) | 105 (89–127) | 92 (78–116) | 135 (115–148) | 126 (73–159) | 0.024[b] |

(*Continued*)

**Table 4.** (Continued)

| Parameters | Reference range | Day 0 | Day 1 | Day 3 | Day 7 | Day 14 | P value |
|---|---|---|---|---|---|---|---|
| SOFA >10 | | 85 (56–107) | 106 (51–135) | 91 (61–118) | 82 (63–129)[#] | 96 (67–116) | 0.121[c] |

Values represent median (IQR). SOFA: sequential organ failure assessment score. P values were calculated with the use of generalized estimating equations (GEE): (a): time effect, (b): group effect and (c): time-group interaction. Pairwise comparisons significant at the 0.05 level:

(*): time effect—pooled patients: each time point vs. Day 0.

(#): between group comparisons (group SOFA>10 vs. group SOFA ≤10) at each time point.

The hemostatic disorders observed in critically ill COVID-19 patients are complex and multifactorial. The combination of varying degrees of hypoxemia, immune-mediated endothelial damage/dysfunction, and systemic inflammation have been implicated in the physiopathology of hypercoagulability state observed in COVID-19 patients [29]. Therefore, several authors have demonstrated that COVID-19 infection is associated with a high rate of venous and arterial thrombotic manifestations, such as DVT, pulmonary embolism, catheter-related thrombosis and ischemic stroke [5, 12, 30, 31]. Therefore, caution for thromboembolic events should be advised especially in the most severe patients, and the strategy for DVT prophylaxis / anticoagulation individualized.

Panigada and cols. recently demonstrated that patients admitted to the ICU infected with COVID-19 exhibited hypercoagulability by thromboelastography [8]. Similarly, we found that most patients admitted to the ICU with severe SARS-CoV-2 infection exhibited a pronounced hypercoagulability state, characterized by increased ROTEM INTEM, EXTEM and FIBTEM

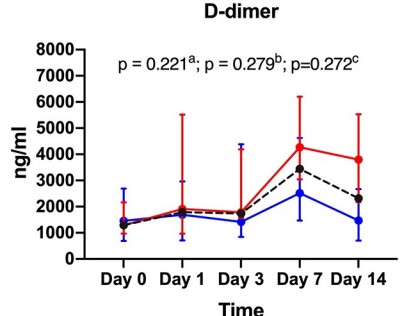
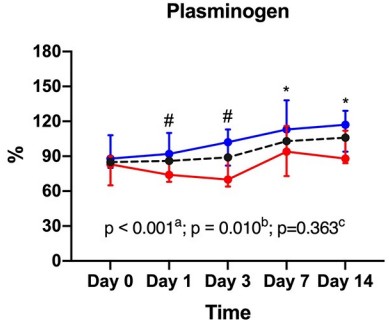
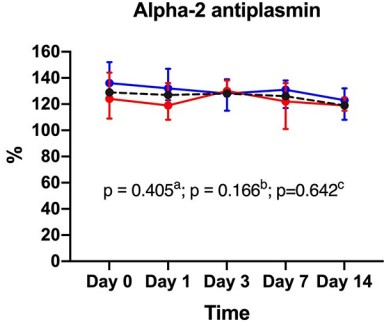
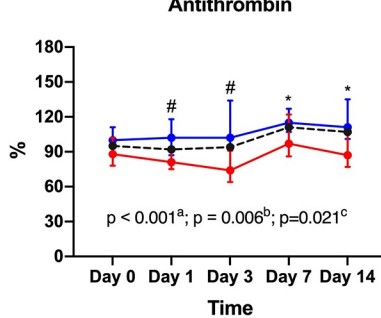
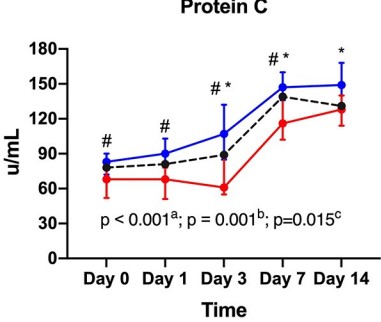
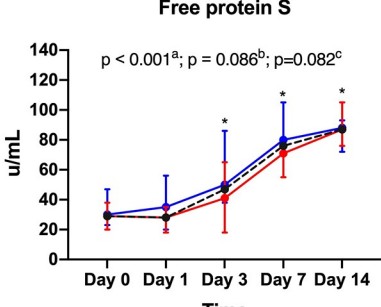

**Fig 2. Fibrinolysis and endogenous inhibitors of coagulation.**

maximum clot firmness. Nevertheless, we did not observe consumption coagulopathy and laboratory findings suggesting DIC as observed by Tang and cols [11]. Our results are in line with other studies that did not show a DIC state in severe critical ill COVID-19 patients [5, 8, 30].

Hypercoagulability may be related to augmented levels of pro-coagulant factors, reduced levels of the naturally occurring anti-coagulant factors or both. Fibrinogen levels, which were initially high in both groups at ICU admission, decreased during the ICU stay. The decrease was more pronounced in patients with lower maximum SOFA score (group SOFA ≤ 10) and after day 7. Also, the naturally occurring anti-coagulant factors, such as protein C and anti-thrombin were lower in the sicker patients (SOFA >10). The differences observed over time between the groups in our study may reflect a higher pro-coagulant tendency in the higher SOFA group and may explain the worst clinical outcomes observed in the group of patients with SOFA > 10.

Protein C has anti-coagulant, profibrinolytic, and anti-inflammatory effects [33]. It has been shown that septic patients with low plasma protein C concentrations have a higher incidence of organ dysfunction and worse outcomes [33–35]. Furthermore, the occurrence of fibrinolysis suppression in patients with septic shock was associated with disease severity and lower survival [36]. Additionally, it has been shown that patients with acute respiratory distress syndrome (ARDS) have higher levels of plasminogen activator inhibitor-1 (PAI-1), which contribute to fibrin deposition in lung parenchyma [37]. This abnormality has been shown also during the SARS-CoV-1 pandemic [38].

There are some pathophysiological changes secondary to COVID-19 infection that may contribute to a greater chance of patients infected with COVID-19 to develop thrombotic complications as an increased angiotensin II expression secondary to angiotensin-converting enzyme 2 receptor binding and consequently augmented plasminogen activator inhibitor C-1 expression with a reduced fibrinolysis in the anticoagulation system [39].

Further, angiotensin II–mediated pulmonary vasoconstriction can predispose to stasis and hypercoagulability, as can COVID-19 induction of antiphospholipid antibodies and complement during cytokine storms, causing vasculitis and microthromboses [40].

Patients in group SOFA > 10 had lower plasminogen levels than patients with SOFA <10, which may represent less fibrinolysis activity during a state of hypercoagulability and, consequently, a greater probability of microthrombi formation in the microcirculation of different organs. This might explain the higher rate of organ dysfunction in group SOFA > 10. Nevertheless, our data do not allow us to confirm or refute this hypothesis. We did not evidence differences in the fibrinolytic behavior in both groups accordingly to ROTEM. One possible explanation for this fact is the high values of fibrinogen present in both groups, which compromise detection of increased thrombotic activity by ROTEM maximum lysis. The hypothesis that the fibrinolytic system is overwhelmed during COVID-19 infection has been commented in a recent paper discussing the association of COVD-19 and the fibrinolytic pathway [7].

We believe that despite several studies addressing the alterations in the coagulation system of COVID-19 patients, [41–46] our study continues to bring important news because it was one of the few that evaluated the behavior of the coagulation system, of critically critical patients infected with COVID-19, during two weeks of hospitalization. Also, our study was one of the few where we tried to assess the impact of the degree of change in the coagulation system and its impact on the evolution of organ dysfunctions presented by patients during the two weeks of study.

Several mechanics can explain the relationship between viral infection and our findings, as endothelial cell disruption, tissue factor expression, and activation of the coagulation cascade by cytokines released during viral infections are other possible mechanisms of thrombosis. This pro-inflammatory state can promote microthrombosis in the vascular lung system and

consequently promoting more hypoxia with local impact creating a deleterious positive thrombo-inflammatory feedback loop [39, 47, 48].

Our study has some limitations. First, it is a single-center study with a relatively small sample size. However, to the best of our knowledge, it was the first study to perform a comprehensive and longitudinal assessment of coagulation profile in critically ill COVID-19 patients during the first two weeks of ICU stay. Secondly, during the study period, all the ICU beds available in our department were destinated to COVID-19 patients. Thus, inclusion of a control group without severe SARS-CoV-2 infection was not possible. Nevertheless, the coagulation profile of patients admitted to the ICU in our center has been recently addressed [15]. Also, a recent study has already demonstrated a higher incidence of thrombotic complications were diagnosed in COVID-19 ARDS patients than in patients with non-COVID-19 ARDS [5]. Third, all patients were already receiving anticoagulants as DVT prophylaxis or systemic anticoagulation and these could change the ROTEM results. Fourth, we did not evaluate all factors involved in fibrinolysis, such as PAI-1 and plasmin, which preclude us to fully understand the role of fibrinolytic system on COVID-19 induced coagulopathy and also other laboratory tests as prothrombin fragment 1+2, thrombin-anti-thrombin complexes and endogenous thrombin potential assays were not done to better understand the hypercoagulability state of such patients. Finally, we not measured any marker for endothelial dysfunction which probably contribute to the modifications on the coagulation system of the severe patients infected with SARS-CoV-2. Nevertheless, our study was the first to demonstrate some aspects of coagulation disorders that can occur in critical patients infected by COVID-19, especially the deficiency of naturally anti-coagulant factors.

## Conclusion

Patients admitted to the ICU with COVID-19 have a pronounced hypercoagulability state, characterized by impaired endogenous anticoagulation system and decreased fibrinolysis. Moreover, the magnitude of coagulation abnormalities seems to correlate with the severity of organ dysfunction. The hypercoagulability state of patients infected with SARS-CoV-2 was detected by ROTEM and other coagulation tests but not with usual coagulation tests. Our findings highlight the role of rotational thromboelastometry when monitoring the coagulation system in ICU patients with COVID-19 and also demonstrated that the mechanisms to explain the hypercoagulability state of patients infected with SARS-CoV-2 is very complex and need more studies.

## Supporting information

**S1 Table. Fibrinolysis and endogenous inhibitors of coagulation.**
(DOCX)

## Acknowledgments

We thank Helena Spalic for proofreading this manuscript.

## Author Contributions

**Conceptualization:** Thiago Domingos Corrêa, Ricardo Luiz Cordioli, João Carlos Campos Guerra, Valdir Fernandes de Aranda.

**Data curation:** Bruno Caldin da Silva.

**Formal analysis:** Bruno Caldin da Silva, Roseny dos Reis Rodrigues.

**Investigation:** Thiago Domingos Corrêa, Ricardo Luiz Cordioli, João Carlos Campos Guerra, Guilherme Martins de Souza, Thais Dias Midega, Niklas Söderberg Campos, Bárbara Vieira Carneiro, Flávia Nunes Dias Campos, Hélio Penna Guimarães, Gustavo Faissol Janot de Matos.

**Methodology:** Thiago Domingos Corrêa, Roseny dos Reis Rodrigues, Gustavo Faissol Janot de Matos, Valdir Fernandes de Aranda.

**Resources:** João Carlos Campos Guerra, Valdir Fernandes de Aranda, Leonardo José Rolim Ferraz.

**Supervision:** Thiago Domingos Corrêa, Ricardo Luiz Cordioli.

**Visualization:** Roseny dos Reis Rodrigues.

**Writing – original draft:** Bruno Caldin da Silva.

**Writing – review & editing:** Thiago Domingos Corrêa, Ricardo Luiz Cordioli.

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
