## [Decision Letter · Decision Letter 0]

17 Sep 2020

PONE-D-20-25608

COAGULATION PROFILE OF COVID-19 PATIENTS ADMITTED TO THE ICU: AN EXPLORATORY STUDY

PLOS ONE

Dear Dr. Cordioli,

Thank you for submitting your manuscript to PLOS ONE. After careful consideration, we feel that it has merit but does not fully meet PLOS ONE’s publication criteria as it currently stands. Therefore, we invite you to submit a revised version of the manuscript that addresses the points raised during the review process.

The study has been revised by two experts in the field. They note several concerns that should be adequatedly considered. The study does not have a group of ICU patients to look for specific factors altered in COVID-19. The authors should provide convincing published evidence in order to understand in which way the altered parameters are specific of COVID-19. Furhter, heparin could affect ROTEM measurements. Finally, protein S values are reported as free protein S values. This should be indicated in the table and description of the results. The total values of protein S would be essential to understand the type of deficiency associated with COVID19. If possible protein S activity values would be interesting to know. The authors should try to update their discussion with the latest research on this very active field of research.

We look forward to receiving your revised manuscript.

Kind regards,

Pablo Garcia de Frutos

Academic Editor

PLOS ONE

Journal Requirements:

2. In your Methods section, please provide additional information about the participant recruitment method and the demographic details of your participants. Please ensure you have provided sufficient details to replicate the analyses such as:

a) the recruitment date range (month and year),

b) a statement as to whether your sample can be considered representative of a larger population, and

c) a description of how participants were recruited.

Reviewers' comments:

Reviewer's Responses to Questions

**Comments to the Author**

1. Is the manuscript technically sound, and do the data support the conclusions?

Reviewer #1: Yes

Reviewer #2: Yes

2. Has the statistical analysis been performed appropriately and rigorously? 

Reviewer #1: Yes

Reviewer #2: Yes

3. Have the authors made all data underlying the findings in their manuscript fully available?

Reviewer #1: No

Reviewer #2: Yes

4. Is the manuscript presented in an intelligible fashion and written in standard English?

Reviewer #1: Yes

Reviewer #2: Yes

5. Review Comments to the Author

Reviewer #1: The authors have measured coagulation tests serially in the intensive care unit in 30 COVID19 patients. The authors found hypercoagulability. A main problem is that there is no control group with non-COVID ICU patients. We will therefor not be able to know whether ICU treated COVID19 patients have higher coagulation activation than other ICU patients. Still of course the longitudinal measurement is of interest.

Comments

1. No non-COVID19 ICU control group

2. I number of patients have been treated with UFH ie heparin. This may affect the ROTEM test.

3. It would have been nice to have a figures where the individuals patients values are plotted over time for the most important analysis at least.

4. In table 3 and 4 it is unclear what the p-values actually is testing.

5. There is a number of papers already published on this topic. What is the novelty with the present paper? For instance: a) Hardy M, Douxfils J, Bareille M, Lessire S, Gouin-Thibault I, Fontana P, Lecompte T, Mullier F. Studies on hemostasis in COVID-19 deserve careful reporting of the laboratory methods, their significance and their limitations. J Thromb Haemost. 2020 Aug 13:10.1111/jth.15061. b) Collett LW, Gluck S, Strickland RM, Reddi BJ. Evaluation of coagulation tatus using viscoelastic testing in intensive care patients with coronavirus disease 2019 (COVID-19): An observational point prevalence cohort study. Aust Crit Care. 2020 Jul 21:S1036-7314(20)30254-X. d) Creel-Bulos C, Auld SC, Caridi-Scheible M, Barker N, Friend S, Gaddh M, Kempton CL, Maier C, Nahab F, Sniecinski R. Fibrinolysis Shutdown and Thrombosis in A COVID-19 ICU. Shock. 2020 Aug 4. C) Ibañez C, Perdomo J, Calvo A, Ferrando C, Reverter JC, Tassies D, Blasi A. High D dimers and low global fibrinolysis coexist in COVID19 patients: what is going on in there? J Thromb Thrombolysis. 2020 Jul 15:1–5. D) Nougier C, Benoit R, Simon M, Desmurs-Clavel H, Marcotte G, Argaud L, David JS, Bonnet A, Negrier C, Dargaud Y. Hypofibrinolytic state and high thrombin generation may play a major role in SARS-COV2 associated thrombosis. J Thromb Haemost. 2020 Jul 15:10.1111/jth.15016. doi: 10.1111/jth.15016. E)Pavoni V, Gianesello L, Pazzi M, Stera C, Meconi T, Frigieri FC. Evaluation of coagulation function by rotation thromboelastometry in critically ill patients with severe COVID-19 pneumonia. J Thromb Thrombolysis. 2020

Reviewer #2: This is an intriguing study focused on an important topic linked to COVID-19.

The relevance of coagulation abnormalities in COVID-19 patients is clearly underscored by the relationship between degree of severity of disease and indicators of abnormal clotting.

It would be interesting to more clearly postulate in the discussion the mechanism by which the viral infection may lead to the reported findings.

Furthermore, the role of Vitamin K abnormalities in patients with defective ROTEM should be discussed.

Future investigations should attempt to validate the current observation in independent patients’ cohorts. Finally, conducting a long-term follow up on patients with ROTEM severe abnormalities may provide further insights into the long-term vascular complications of the disease.

6. PLOS authors have the option to publish the peer review history of their article (what does this mean?). If published, this will include your full peer review and any attached files.

Reviewer #1: No

Reviewer #2: No

---

## [Author Response · Author response to Decision Letter 0]

26 Oct 2020

Dear Editor and Reviewers

Thank you very much for the interesting and important inputs to our study

PONE-D-20-25608 - COAGULATION PROFILE OF COVID-19 PATIENTS ADMITTED TO THE ICU: AN EXPLORATORY STUDY

Responses to the Editor

The study does not have a group of ICU patients to look for specific factors altered in COVID-19. 

We didn’t do a control group at the time of the study, as the study planning and even data collection occurred at the beginning of the pandemic in Brazil and the Hospital Israelita Albert Einstein, where the study was done, was reorganized to treat, almost exclusively, patients infected with COVID-19 as it happened around the world. We addressed such limitation in our text “Secondly, during the study period, all the ICU beds available in our department were designated to COVID-19 patients. Thus, inclusion of a control group without severe SARS-CoV-2 infection was not possible. Nevertheless, the coagulation profile of patients admitted to the ICU in our center has been recently addressed [15]” The reference cited above (#15) evaluated 531 critical ill patients and showed normal ROTEM profille in the majority of ICU, non COVID-19, patients. 

Also, to make it clearer, we included this sentence following the previous paragraph “Also, a recent study have already demonstrated a higher incidence of thrombotic complications were diagnosed in COVID-19 ARDS patients than in patients with non-COVID-19 ARDS”. (included reference number 5 already cited in our study)

So, it was not possible to include a control group during the study period, only on July the pandemic started to be more controlled in our city and we returned to have non-COVID-19 patients in our ICU. By the reasons discussed, it would have taken a long time to have had a control group and then delay the manuscript submission.

The authors should provide convincing published evidence in order to understand in which way the altered parameters are specific of COVID-19

We agree that there are indirect effects of any infection, such as through severe illness with subsequently inflammatory response that may predispose patients to thrombotic events and this is not specific for COVID-19 patients as well as hypoxia can contribute to thrombotic events and again it is not specific for hypoxemic COVID-19 patients. However, COVID-19 patients demonstrated a higher incidence of thrombotic events that highlighted the importance of better understand its specific changes in the coagulation system and it is the main objective of our study. The following epidemiological studies highlighted the greater incidence of thrombosis events in patients hospitalized with COVID-19 specially in ICU patients 

1. Klok FA, Kruip M, van der Meer NJM, Arbous MS, Gommers D, Kant KM, et al. Incidence of thrombotic complications in critically ill ICU patients with COVID-19. Thrombosis research. 2020;191:145-7.

2. Helms J, Tacquard C, Severac F, Leonard-Lorant I, Ohana M, Delabranche X, et al. High risk of thrombosis in patients with severe SARS-CoV-2 infection: a multicenter prospective cohort study. Intensive care medicine. 2020;46(6):1089-98.

3. Lodigiani C, Iapichino G, Carenzo L, Cecconi M, Ferrazzi P, Sebastian T, et al. Venous and arterial thromboembolic complications in COVID-19 patients admitted to an academic hospital in Milan, Italy. Thrombosis research. 2020;191:9-14.

4. Bikdeli B, Madhavan MV, Jimenez D, Chuich T, Dreyfus I, Driggin E, et al. COVID-19 and Thrombotic or Thromboembolic Disease: Implications for Prevention, Antithrombotic Therapy, and Follow-Up: JACC State-of-the-Art Review. Journal of the American College of Cardiology. 2020;75(23):2950-73.

5. Klok FA, et al. .Confirmation of the high cumulative incidence of thrombotic complications in critically ill ICU patients with COVID-19: An updated analysis. Thromb Res. 2020. PMID: 32381264 

We included the further sentence, on the Discussion section, to address the correlation of COVID-19 and thrombosis profile of patients with COVID-19: “There are some pathophysiological changes secondary to COVID-19 infection that may contribute to a greater chance of patients infected with COVID-19 to develop thrombotic complications as an increased angiotensin II expression secondary to angiotensin-converting enzyme 2 receptor binding and consequently augmented plasminogen activator inhibitor C-1 expression with a reduced fibrinolysis in the anticoagulation system. (reference included: Dolhnikoff M, Duarte-Neto AN, de Almeida Monteiro RA, et al. Pathological evidence of pulmonary thrombotic phenomena in severe COVID-19. Thromb Haemost 2020 Jun;18(6):1517-1519.)

 Further, angiotensin II–mediated pulmonary vasoconstriction can predispose to stasis and hypercoagulability, as can COVID-19 induction of antiphospholipid antibodies and complement during cytokine storms, causing vasculitis and microthromboses. (reference included: Medcalf RL, Keragala CB, Myles PS. Fibrinolysis and COVID-19: a plasmin paradox. J Thromb Haemost. 2020.)

Heparin could affect ROTEM measurements

We agree with this statement and it is one of the reasons that we changed in the Discussion section where it was included on the limitation paragraph: Third, all patients were already receiving anticoagulants as DVT prophylaxis or systemic anticoagulation and these could change the ROTEM results.

Concerning the comments regarding protein S:

We changed on Material and Methods section at the Laboratory analysis subsection “Fibrinolysis and endogenous anticoagulation system” that we measured the free protein S and in the Table and when it was cited in the text we changed in the manuscript and always wrote “free protein S” to be more exact. 

We understand your comment about the importance of total protein S but protein S is a vitamin K-dependent glycoprotein, which acts as a cofactor for Activated Protein C, increasing its anticoagulant and profibrinolytic effects. Protein S is present in plasma in two forms: free protein S (40%) and protein S bound to the transport protein of the complement C4b fraction (60%). The two forms are in dynamic equilibrium and only free Protein S has biological activity. These are the reasons that we decided to include free protein S since its biological activity.

References to our decision: 

1. Suzuki K, Nishioka J. Plasma Protein S Activity measured using Protac, a Snake Venom Derived Activator of Protein C, Thromb. Res. 1988; 49: 214-251.

2. Faioni EM, Valsecchi C, Palla A, Taioli E, Razzari C, Mannucci PM. Free protein S deficiency is a Risk Factor for Venous Thrombosis. Thromb Haemost 1997; 78: 1343-1346

Response to Journal Requirements:

When submitting your revision, we need you to address these additional requirements. Please ensure that your manuscript meets PLOS ONE's style requirements, including those for file naming. 

 We did the changed above according to PLOS ONE's style requirements

In your Methods section, please provide additional information about the participant recruitment method and the demographic details of your participants. Please ensure you have provided sufficient details to replicate the analyses such as:

a) the recruitment date range (month and year),

b) a statement as to whether your sample can be considered representative of a larger population, and

c) a description of how participants were recruited.

 We included the paragraph on the Methods Section: “Participants were recruited between March 29, 2020 through May 13, 2020 and they could represent the majority of severe patients infected by COVID-19, once there were few exclusion criteria and the participants were recruited with waiver of informed consent once it was an observational study without any intervention and consecutive patients admitted in the Intensive Care Unit were recruited following the inclusion and exclusion criteria until we completed 30 patients included in the study.”

Response to the comments of the reviewers 

Have the authors made all data underlying the findings in their manuscript fully available?

 Reviewer #1: No. 

We provided now all our dataset, in the datadryad.org following the Plos One policy after your comment.

Please, find bellow the message from datadryad: “Your dataset has been assigned a unique identifier, called a DOI (doi:10.5061/dryad.pg4f4qrn3). If your dataset is associated with a manuscript submission, you may provide this DOI to the journal, although it will not be live until the dataset is published. For private access prior to publication, you may share your dataset using this temporary link: https://datadryad.org/stash/share/iz0FlU7yXkYg4_Yh2PM_2y0cPIPTyiD6_lry0NMfJKM.”

Review Comments to the Author

Comments

No non-COVID19 ICU control group

We agree that it should be interest to have a control group of ICU non-COVID-19 patients, but we didn’t do a control group at the time of the study, as the study planning and even data collection occurred at the beginning of the pandemic in Brazil and the Hospital Israelita Albert Einstein, where the study was done, was reorganized to treat, almost exclusively, patients infected with COVID-19 as it happened around the world. We addressed such limitation in our text “Secondly, during the study period, all the ICU beds available in our department were destinated to COVID-19 patients. Thus, inclusion of a control group without severe SARS-CoV-2 infection was not possible. Nevertheless, the coagulation profile of patients admitted to the ICU in our center has been recently addressed [15]” 

The reference cited above (#15) evaluated 531 critical ill patients and showed normal ROTEM profilte in the majority of ICU, non COVID-19, patients. 

Also, to make it clearer, we included this sentence following the previous paragraph “Also, a recent study have already demonstrated a higher incidence of thrombotic complications were diagnosed in COVID-19 ARDS patients than in patients with non-COVID-19 ARDS”. (included refence number 5 already cited in our study)

So, it was not possible to include a control group during the study period, only on July the pandemic started to be more controlled in our city and we returned to have non-COVID-19 patients in our ICU. By the reasons discussed, it would have taken a long time to have had a control group and then delay the submission

I number of patients have been treated with UFH ie heparin. This may affect the ROTEM test.

We agree with this statement and it is one of the reasons that we changed in the Discussion section where it was included on the limitation paragraph: Third, all patients were already receiving anticoagulants as DVT prophylaxis or systemic anticoagulation and these could change the ROTEM results. 

Before the pandemic started, the ICU of Hospital Israelita Albert Einstein have already a DVT prophylaxis algorithm and a culture in the institution to introduce drug prophylaxis for DVT in critically ill patients, unless contraindicated, and even more so in patients with COVID-19, due to reports of initial studies already demonstrating a significant incidence of thrombotic events in such patients, then it would take too much time to include patients with COVID-19, whose prophylaxis for DVT was contraindicated, and then the study would take much time to be finished.

It would have been nice to have a figures where the individuals patients values are plotted over time for the most important analysis at least.

We included a figure 

Figure 2. Fibrinolysis and endogenous inhibitors of coagulation. 

Values represent median (IQR). Red lines represent group SOFA >10 (n=14), blue lines represent group SOFA ≤10 (n=16) and black dotted lines represent all patients (n=30). P values were calculated with the use of generalized estimating equations (GEE): (a): time effect, (b): group effect and (c): time-group interaction. Pairwise comparisons significant at the 0.05 level: (*): time effect - pooled patients: each time point vs. Day 0. (#): between group comparisons (group SOFA>10 vs. group SOFA ≤10) at each time point.

And we put the Table 5 as a supplemental material 

In table 3 and 4 it is unclear what the p-values actually is testing.

We included the sentence in the Statistical analysis Section to clarify what p-values actually tested: “To account for longitudinal (repeated measurements) and correlated response continuous variables, between-group differences and within-group differences over time were assessed using generalized estimating equations (GEE), with group (SOFA ≤ 10 and group SOFA > 10) and study time points (time) as predictors. P values for group effect, time effect, and time-group interaction were presented. When a time effect was detected in pooled patients, each time point (Day 1, 3, 7 and 14) was compared against Day 0. When a group effect or a time-group interaction were detected, between group comparisons (group SOFA>10 vs. group SOFA ≤10) were performed at each time point. The Bonferroni method was used to account for multiple comparisons”

Also, we did some modifications on Table 3 to make it clearer and we changed the legend to the following phrase: “Values represent median (IQR). INR: international normalized ratio, aPTT: activated partial thromboplastin time. P values were calculated with the use of generalized estimating equations (GEE): (a): time effect, (b): group effect and (c): time-group interaction. Pairwise comparisons significant at the 0.05 level: (*): time effect - pooled patients: each time point vs. Day 0. (#): between group comparisons (group SOFA>10 vs. group SOFA ≤10) at each time point.”

We also changed the Table 4 as well as its legend to: “Values represent median (IQR). SOFA: sequential organ failure assessment score. P values were calculated with the use of generalized estimating equations (GEE): (a): time effect, (b): group effect and (c): time-group interaction. Pairwise comparisons significant at the 0.05 level: (*): time effect - pooled patients: each time point vs. Day 0. (#): between group comparisons (group SOFA>10 vs. group SOFA ≤10) at each time point.”

We also changed the Table 5, to a supplemental table as well as its legend to: “Values represent median (IQR). SOFA: sequential organ failure assessment score. P values were calculated with the use of generalized estimating equations (GEE): (a): time effect, (b): group effect and (c): time-group interaction. Pairwise comparisons significant at the 0.05 level: (*): time effect - pooled patients: each time point vs. Day 0. (#): between group comparisons (group SOFA>10 vs. group SOFA ≤10) at each time point.” 

There is a number of papers already published on this topic. What is the novelty with the present paper? 

For instance: a) Hardy M, Douxfils J, Bareille M, Lessire S, Gouin-Thibault I, Fontana P, Lecompte T, Mullier F. Studies on hemostasis in COVID-19 deserve careful reporting of the laboratory methods, their significance and their limitations. J Thromb Haemost. 2020 Aug 13:10.1111/jth.15061. 

This is an interesting letter that mainly discussed limits of another study (Nougier C, Benoit R, Simon M, Desmurs-Clavel H, Marcotte G, Argaud L, et al. Hypofibrinolytic state and high thrombin generation may play a major role in sars-cov2 associated thrombosis. J Thromb Haemost. 2020. Online ahead of print. DOI: 10.1111/jth.15016.) showing the complexity in studying the coagulation profile of COVID-19 patients, but our study, different from this letter and also from the study cited above, analyzed evaluated different coagulation parameters on COVID-19 patient and for a longer period.

b) Collett LW, Gluck S, Strickland RM, Reddi BJ. Evaluation of coagulation tatus using viscoelastic testing in intensive care patients with coronavirus disease 2019 (COVID-19): An observational point prevalence cohort study. Aust Crit Care. 2020 Jul 21:S1036-7314(20)30254-X. 

This is an interesting paper, but with a small sample (6 patients) and it evaluated mainly the ROTEM ® profile and also only in a single measurement, our study, on the other hand, evaluated more coagulation examens and also during 14 days

c) Creel-Bulos C, Auld SC, Caridi-Scheible M, Barker N, Friend S, Gaddh M, Kempton CL, Maier C, Nahab F, Sniecinski R. Fibrinolysis Shutdown and Thrombosis in A COVID-19 ICU. Shock. 2020 Aug 4. 

Similar to our study, this study also evaluated 21 patients, and it was demonstrated a fibrinolysis shutdown, but this study assessed mainly the ROTEM® profile and also only in a single day and unlike our study that evaluated different coagulation parameters over time and also compared different clinical groups (SOFA < or > 10).

d) Ibañez C, Perdomo J, Calvo A, Ferrando C, Reverter JC, Tassies D, Blasi A. High D dimers and low global fibrinolysis coexist in COVID19 patients: what is going on in there? J Thromb Thrombolysis. 2020 Jul 15:1–5.

Similar to our study, this study also evaluated 19 patients, and it was also demonstrated a hypofibrinolytic pattern on the viscoelastic examen and elevated D-dimers value as we demonstrated, however this study above did not evaluated other coagulation parameters as we did and we demonstrated a time effect on ROTEM® profile that was not demonstrated in the study above.

e) Nougier C, Benoit R, Simon M, Desmurs-Clavel H, Marcotte G, Argaud L, David JS, Bonnet A, Negrier C, Dargaud Y. Hypofibrinolytic state and high thrombin generation may play a major role in SARS-COV2 associated thrombosis. J Thromb Haemost. 2020 Jul 15:10.1111/jth.15016. doi: 10.1111/jth.15016. 

It is a very interesting paper which the authors investigated several different coagulation parameters, however the patients included were less sicker with a lower mean SOFA score, lower rate of mechanical ventilation and also it was only done discussed one single day measurement of the coagulation profile, so it was not studied the coagulation behavior along two weeks as we done and our study the patients included were general more sicker.

f)Pavoni V, Gianesello L, Pazzi M, Stera C, Meconi T, Frigieri FC. Evaluation of coagulation function by rotation thromboelastometry in critically ill patients with severe COVID-19 pneumonia. J Thromb Thrombolysis. 2020

This study is very interesting, since as our study, the authors investigated the coagulation profile over 10 days (we investigated during two weeks) but the study above include less severe patients, where only 10% of the patients need mechanical ventilation, the mean SOFA was less than the less critical group form our study (group SOFA < 10), and there are not two groups as our study to evaluate the impact and relationship of the coagulation system disorders and the organ dysfunction in ICU COVID-19 patients, one issue that our study has addressed.

Thank you for sharing these important and interesting references. 

We included in our Discussion section the following sentence: “We believe that despite several studies addressing the alterations in the coagulation system of COVID-19 patients, (included all the references above) our study continues to bring important news because it was one of the few that evaluated the behavior of the coagulation system, of critically critical patients infected with COVID-19, during two weeks of hospitalization. Also, our study was one of the few where we tried to assess the impact of the degree of change in the coagulation system and its impact on the evolution of organ dysfunctions presented by patients during the two weeks of study”. 

Reviewer #2: This is an intriguing study focused on an important topic linked to COVID-19.

The relevance of coagulation abnormalities in COVID-19 patients is clearly underscored by the relationship between degree of severity of disease and indicators of abnormal clotting.

It would be interesting to more clearly postulate in the discussion the mechanism by which the viral infection may lead to the reported findings.

We included this sentence on the Discussion section to address this relationship: “Several mechanics can explain the relationship between viral infection and our findings, as endothelial cell disruption, tissue factor expression, and activation of the coagulation cascade by cytokines released during viral infections are other possible mechanisms of thrombosis. This pro-inflammatory state can promote microthrombosis in the vascular lung system and consequently promoting more hypoxia with local impact creating a deleterious positive thromboinflammatory feedback loop (reference included: Dolhnikoff M, Duarte-Neto AN, de Almeida Monteiro RA, et al. Pathological evidence of pulmonary thrombotic phenomena in severe COVID-19. Thromb Haemost 2020 Jun;18(6):1517-1519. AND Wool GD, Miller JL. The Impact of COVID-19 Disease on Platelets and Coagulation Pathobiology. 2020 Oct 13:1-13. doi: 10.1159/000512007. AND Escher R, Breakey N, Lämmle B. Severe COVID-19 infection associated with endothelial activation. Thromb Res. 2020 Jun;190:62.)

 Furthermore, the role of Vitamin K abnormalities in patients with defective ROTEM should be discussed.

We agree with the sentence above and is one of the reason that our exclusion criteria were, as cited in the manuscript: “Exclusion criteria included pregnancy, previous known coagulopathy, currently use of systemic anticoagulants or anti-platelet therapy or vitamin K antagonists, moribund patients and patients who presented cardiac arrests. 

Future investigations should attempt to validate the current observation in independent patients’ cohorts. 

We completely agree that it should be very interesting and important to validate the current observation in a ICU non COVID-19 population, however as explained in sentences above it would have taken a long time to have this population and we did not know when the peak of the pandemic in Brazil would be better allowing to return to a more stable situation where critical ill patients without COVID-19 infection would be hospitalized and we would be able to evaluate this population regarding deeply the coagulation system as we did in our study with severe ill COVID-19 patients. We intend to do this in a future research

Finally, conducting a long-term follow up on patients with ROTEM severe abnormalities may provide further insights into the long-term vascular complications of the disease. 

We completely agree with this idea, and maybe we are going to evaluate for a longer period the coagulation system of critical ill ICU COVID-19 patients that stay more than 01 month in the ICU and/or hospital.

Thank you for your important inputs

Sincerely, 

Ricardo Luiz Cordioli

26TH, October, 2020

---

## [Decision Letter · Decision Letter 1]

18 Nov 2020

PONE-D-20-25608R1

COAGULATION PROFILE OF COVID-19 PATIENTS ADMITTED TO THE ICU: AN EXPLORATORY STUDY

PLOS ONE

Dear Dr. Cordioli,

Thank you for submitting your manuscript to PLOS ONE. After careful consideration, we feel that it has merit but does not fully meet PLOS ONE’s publication criteria as it currently stands. Therefore, we invite you to submit a revised version of the manuscript that addresses the points raised during the review process.

The reviewers consider that their initial concerns have been addressed. One reviewers suggests some minor changes that should be considered by the authors, eitherr by changing their text or by providing a reasoned answer.

We look forward to receiving your revised manuscript.

Kind regards,

Pablo Garcia de Frutos

Academic Editor

PLOS ONE

Reviewers' comments:

Reviewer's Responses to Questions

**Comments to the Author**

1. If the authors have adequately addressed your comments raised in a previous round of review and you feel that this manuscript is now acceptable for publication, you may indicate that here to bypass the “Comments to the Author” section, enter your conflict of interest statement in the “Confidential to Editor” section, and submit your "Accept" recommendation.

Reviewer #1: All comments have been addressed

Reviewer #2: All comments have been addressed

2. Is the manuscript technically sound, and do the data support the conclusions?

Reviewer #1: Yes

Reviewer #2: Yes

3. Has the statistical analysis been performed appropriately and rigorously? 

Reviewer #1: Yes

Reviewer #2: (No Response)

4. Have the authors made all data underlying the findings in their manuscript fully available?

Reviewer #1: Yes

Reviewer #2: Yes

5. Is the manuscript presented in an intelligible fashion and written in standard English?

Reviewer #1: Yes

Reviewer #2: Yes

6. Review Comments to the Author

Reviewer #1: Thea authors have changed the paper according to reviewer suggestions. I have only a few minor but important comments.

1. The authors do not present in the abstract the increased d-dimer levels nor the transient decreased free protein S levels and plasminogen levels.

2. I do not agree with this sentence in the abstract ¨The hypercoagulability state of COVID-19 patients was only detected by ROTEM¨ or the sentences in discussion or conclusions:¨ Finally, the hypercoagulability state of severe COVID-19 patients was detected by ROTEM, while conventional coagulation tests remained unchanged.¨ ¨The hypercoagulability state of patients infected with SARS-CoV-2 was detected by ROTEM but not with conventional coagulation tests.¨ Ordinary coagulation analysis showed increased fibrinogen, increased d-dimer, slightly increased antiplasmin, transient decreased free protein S. and slightly transient decreased plasminogen. Thus, not only ROTEM could identify a hypercoagulable state in COVID infection.

3. A limitation is that the authors have not measured markers of hypercoagulability such as F1+2, TAT and endogenous thrombin potential. Nor have the authors measured any marker for endothelial dysfunction. I think this should be stated among limitations in the discussion. This has not only relevance for the authors conclusion that the hypercoagulability of COVID19 may be detected only by ROTEM and not conventional coagulation tests but also regarding the cause of hypercoagulability.

Reviewer #2: No further comments .................................................................................................................................

7. PLOS authors have the option to publish the peer review history of their article (what does this mean?). If published, this will include your full peer review and any attached files.

Reviewer #1: No

Reviewer #2: No

---

## [Author Response · Author response to Decision Letter 1]

19 Nov 2020

Dear Editor and Reviewers

Thank you very much for the interesting and important inputs to our study

PONE-D-20-25608 - COAGULATION PROFILE OF COVID-19 PATIENTS ADMITTED TO THE ICU: AN EXPLORATORY STUDY

Response to the comments of the reviewers 

1. The authors do not present in the abstract the increased d-dimer levels nor the transient decreased free protein S levels and plasminogen levels.

We changed the result paragraph on the Abstract Section:

“Thirty patients were studied. Some conventional coagulation tests, as aPTT, PT and INR remained unchanged during the study period, while alterations on others coagulation laboratory tests were detected. Fibrinogen levels were increased in both groups. ROTEM maximum clot firmness increased in both groups from Day 0 to Day 14. Moreover, ROTEM – FIBTEM maximum clot firmness was high in both groups, with a slight decrease from day 0 to day 14 in group SOFA ≤ 10 and a slight increase during the same period in group SOFA > 10. Fibrinolysis was low and decreased over time in all groups, with the most pronounced decrease observed in INTEM maximum lysis in group SOFA > 10. Also, D-dimer plasma levels were higher than normal reference range in both groups and free protein S plasma levels were low in both groups at baseline and increased over time, Finally, patients in group SOFA > 10 had lower plasminogen levels and Protein C than patients with SOFA <10, which may represent less fibrinolysis activity during a state of hypercoagulability.” 

2. I do not agree with this sentence in the abstract ¨The hypercoagulability state of COVID-19 patients was only detected by ROTEM¨ or the sentences in discussion or conclusions:¨Finally, the hypercoagulability state of severe COVID-19 patients was detected by ROTEM, while conventional coagulation tests remained unchanged.¨ ¨The hypercoagulability state of patients infected with SARS-CoV-2 was detected by ROTEM but not with conventional coagulation tests.¨ Ordinary coagulation analysis showed increased fibrinogen, increased d-dimer, slightly increased antiplasmin, transient decreased free protein S. and slightly transient decreased plasminogen. Thus, not only ROTEM could identify a hypercoagulable state in COVID infection.

We changed the conclusion paragraph on the Abstract Section:

“The hypercoagulability state of COVID-19 patients was not only detected by ROTEM but it much more complex, where changes were observed on the fibrinolytic and endogenous anticoagulation system.”

We changed the paragraph on the Discussion Section:

“Finally, the hypercoagulability state of severe COVID-19 patients was detected by ROTEM and modifications on some coagulation tests related to the fibrinolytic and endogenous anticoagulation system, while the more common conventional coagulation tests, as aPTT (sec) INR and platelets, remained unchanged.” 

We changed the paragraph on the Conclusion Section:

“The hypercoagulability state of patients infected with SARS-CoV-2 was detected by ROTEM and other coagulation tests but not with usual coagulation tests. Our findings highlight the role of rotational thromboelastometry when monitoring the coagulation system in ICU patients with COVID-19 and also demonstrated that the mechanisms to explain the hypercoagulability state of patients infected with SARS-CoV-2 is very complex and need more studies”.

3. A limitation is that the authors have not measured markers of hypercoagulability such as F1+2, TAT and endogenous thrombin potential. Nor have the authors measured any marker for endothelial dysfunction. I think this should be stated among limitations in the discussion. This has not only relevance for the authors conclusion that the hypercoagulability of COVID19 may be detected only by ROTEM and not conventional coagulation tests but also regarding the cause of hypercoagulability.

We agree with your important comment and we included in the limitation paragraph: 

“Fourth, we did not evaluate all factors involved in fibrinolysis, such as PAI-1 and plasmin, which preclude us to fully understand the role of fibrinolytic system on COVID-19 induced coagulopathy and also other laboratory tests as prothrombin fragment 1+2, thrombin-anti-thrombin complexes and endogenous thrombin potential assays were not done to better understand the hypercoagulability state of such patients. Finally, we not measured any marker for endothelial dysfunction which probably contribute to the modifications on the coagulation system of the severe patients infected with SARS-CoV-2. Nevertheless, our study was the first to demonstrate some aspects of coagulation disorders that can occur in critical patients infected by COVID-19, especially the deficiency of naturally anti-coagulant factors.”

Thank you for your important inputs

Sincerely, 

Ricardo Luiz Cordioli

19TH, November, 2020

---

## [Editor Report · Decision Letter 2]

25 Nov 2020

COAGULATION PROFILE OF COVID-19 PATIENTS ADMITTED TO THE ICU: AN EXPLORATORY STUDY

PONE-D-20-25608R2

Dear Dr. Cordioli,

We’re pleased to inform you that your manuscript has been judged scientifically suitable for publication and will be formally accepted for publication once it meets all outstanding technical requirements.

Kind regards,

Pablo Garcia de Frutos

Academic Editor

PLOS ONE
---

## [Editor Report · Acceptance letter]

1 Dec 2020

PONE-D-20-25608R2 

COAGULATION PROFILE OF COVID-19 PATIENTS ADMITTED TO THE ICU: AN EXPLORATORY STUDY 

Dear Dr. Cordioli:

I'm pleased to inform you that your manuscript has been deemed suitable for publication in PLOS ONE. Congratulations! Your manuscript is now with our production department. 

Kind regards, 

on behalf of

Dr. Pablo Garcia de Frutos 

Academic Editor

PLOS ONE